# PIM1 Inhibition Affects Glioblastoma Stem Cell Behavior and Kills Glioblastoma Stem-like Cells

**DOI:** 10.3390/ijms222011126

**Published:** 2021-10-15

**Authors:** Carolin Seifert, Ellen Balz, Susann Herzog, Anna Korolev, Sebastian Gaßmann, Heiko Paland, Matthias A. Fink, Markus Grube, Sascha Marx, Gabriele Jedlitschky, Mladen V. Tzvetkov, Bernhard H. Rauch, Henry W. S. Schroeder, Sandra Bien-Möller

**Affiliations:** 1Department of Pharmacology, University Medicine Greifswald, 17489 Greifswald, Germany; carolin.seifert@uni-wuerzburg.de (C.S.); ellen-balz@web.de (E.B.); susannherzog82@gmail.com (S.H.); annako93@web.de (A.K.); gassmann.sebastian2@gmail.com (S.G.); heiko.paland@gmail.com (H.P.); matthias.fink@uni-heidelberg.de (M.A.F.); markus.grube@med.uni-greifswald.de (M.G.); gabriele.jedlitschky@med.uni-greifswald.de (G.J.); mladen.tzvetkov@med.uni-greifswald.de (M.V.T.); bernhard.rauch@uni-oldenburg.de (B.H.R.); 2Department of Neurosurgery, University Medicine Greifswald, 17489 Greifswald, Germany; sascha.marx@uni-greifswald.de (S.M.); henry.schroeder@med.uni-greifswald.de (H.W.S.S.); 3Department of Pharmacology and Toxicology, Carl von Ossietzky University Oldenburg, 26129 Oldenburg, Germany

**Keywords:** glioblastoma, PIM1 kinase, stem-like cells, neurospheres

## Abstract

Despite comprehensive therapy and extensive research, glioblastoma (GBM) still represents the most aggressive brain tumor in adults. Glioma stem cells (GSCs) are thought to play a major role in tumor progression and resistance of GBM cells to radiochemotherapy. The PIM1 kinase has become a focus in cancer research. We have previously demonstrated that PIM1 is involved in survival of GBM cells and in GBM growth in a mouse model. However, little is known about the importance of PIM1 in cancer stem cells. Here, we report on the role of PIM1 in GBM stem cell behavior and killing. PIM1 inhibition negatively regulates the protein expression of the stem cell markers CD133 and Nestin in GBM cells (LN-18, U-87 MG). In contrast, CD44 and the astrocytic differentiation marker GFAP were up-regulated. Furthermore, PIM1 expression was increased in neurospheres as a model of GBM stem-like cells. Treatment of neurospheres with PIM1 inhibitors (TCS PIM1-1, Quercetagetin, and LY294002) diminished the cell viability associated with reduced DNA synthesis rate, increased caspase 3 activity, decreased PCNA protein expression, and reduced neurosphere formation. Our results indicate that PIM1 affects the glioblastoma stem cell behavior, and its inhibition kills glioblastoma stem-like cells, pointing to PIM1 targeting as a potential anti-glioblastoma therapy.

## 1. Introduction

Glioblastoma (GBM) represents the most common brain tumor in adults with no curative therapy currently available. Almost all patients suffering from this highly aggressive tumor get a relapse, and the median survival time is only about 15 months, despite maximal surgical resection, radiation and chemotherapy [1,2]. A specific subpopulation of GBM cells, called glioma stem cells (GSCs), have been associated with recurrence and thus poor outcome. It is assumed that successful treatment of GBM can only be achieved by targeting these highly tumorigenic, pro-mitogenic, pro-angiogenic and therapy-resistant GSCs [3,4]. To date, the regulation of the molecular and biological properties of GSCs has not been well understood and remains elusive. Both natural and synthetic compounds have been evaluated for their potential to modify the behavior of GBM cells and GSCs, for instance, cannabinoids, resveratrol crocetin, the botanical drug PBI-05204, EGFR inhibitors, the PI3K/AKT small-molecule inhibitor A-443654, and many others [5,6,7,8]. So far, the clinical impact of their therapeutic use on patient prognosis is not clear, although many of their modes of action are known.

There is increasing evidence that PIM1, as one of three members of the PIM family, plays a role as an oncogenic kinase. Although these three PIM kinases share more than 60% homology, only PIM1 appears to have a significant impact on tumor formation [9,10]. Initially, the gene encoding PIM1 (Proviral Integration site for Moloney Murine Leukemia Virus) was discovered in murine lymphoma samples and shown to be activatable by the integration of the murine leukemia virus into the 3′untranslated region of the PIM1 gene [11]. The PIM1 gene encodes for two protein isoforms called PIM1S (short isoform, 35 kDa) and PIM1L (long isoform, 45 kDa) which are generated from a single mRNA but using different start codons, AUG and atypical CUG, respectively [12].

In contrast to other kinases, PIM1 is a constitutive active enzyme and does not need to be activated by upstream kinases. In healthy cells, PIM1 expression is rather low but inducible by cytokines, growth factors, or ischemic conditions [13,14]. Besides its role in the pathogenesis of lymphoma, PIM1 expression is also elevated in other cancers. An increased PIM1 expression in stomach, esophagus, bladder, breast and ovarian as well as head and neck cancers has been associated with a worse patient prognosis and resistance to therapy [15,16,17,18,19]. Of note, the use of PIM1 inhibitors or silencing of PIM1 expression in vitro resulted in a significant inhibition of the tumor cell growth through induction of apoptosis, cell cycle arrest and sensitization against chemotherapeutic agents [18,20,21,22].

Recently, our group showed that PIM1 is overexpressed in GBM, and pharmacological inhibition of PIM1 resulted in reduced GBM growth both in vitro and in vivo [23]. Suppression of GBM growth was also observed after U1 adaptor-mediated silencing of PIM1 in GBM cells in a subcutaneous xenograft model in mice [24]. Moreover, inhibition of PIM1 sensitizes GBM cells to apoptosis induced by ABT-737, which targets BCL2, BCL2 like 1 and BCL2 like 2 [25]. Thus, blocking PIM1 may be a new therapeutic option for GBM.

However, to date, the role of PIM1 in the development and maintenance of GSCs has not been described, although PIM1 signaling pathways are associated with cancer stem cells in prostate cancer [26]. Very recently, Jiménez-García and colleagues showed that PIM1 is overexpressed in human breast, uterine and ovarian tumors, correlating with inflammatory features as well as stem cell markers [19]. Moreover, inhibition of PIM1 attenuates the stem cell-like properties of breast cancer cells [27]. Of note, PIM1 interacts with AKT1 signaling [28,29], which has previously been shown to be involved in the behavior of GSCs [6,30,31,32].

Based on these findings and the lack of knowledge on the function of PIM1 in GSCs, we analyzed the role of PIM1 in the stem cell behavior of GBM cells. More specifically, we investigated whether pharmacological inhibition of PIM1 caused alterations in the expression of stem cell and differentiation marker proteins, and measured the expression of PIM1 in parental GBM cells compared with stem-like neurospheres. Furthermore, we analyzed whether stem-like GBM neurospheres cells are disturbed by PIM1 inhibition, to evaluate this strategy as a potential targeted therapy for GBM patients.

## 2. Results

### 2.1. Impact of PIM1 Inhibition on Stem Cell and Differentiation Markers in Human GBM Cells on mRNA Level

First, we analyzed whether inhibition of PIM1 kinase results in an altered expression of candidate stem cell markers (CD44, CD133, Nestin), as well as of the microglia (SPARC) and astrocytic marker protein (GFAP) in the human adherent GBM cell lines LN-18 and U-87 MG on the mRNA level.

Figure 1A (LN-18) and 1B (U-87 MG) show the mRNA data after treatment with either the combined PI3K/PIM1 inhibitor LY294002 (LY) [33], the PIM1 selective blocker quercetagetin (QT) [34] or the PIM1 selective antagonist TCS PIM1-1 (TCS) [35] after 48 h of incubation. We used two different concentrations of the inhibitors, 5 and 50 µM, based on our previous studies, which showed that 5 µM of all tested inhibitors caused only slight blocking of the kinase activity, whereas 50 µM is highly effective as determined by the phosphorylation status of the PIM1 downstream target BAD [23].

On the mRNA level, CD44 and CD133 were not significantly regulated by PIM1 antagonists, with the exception of a 2.64-fold (±1.34) increased CD133 mRNA expression after treatment with 5 µM LY294002 in U-87 MG cells. Regarding Nestin, differences at the mRNA level after treatment with PIM1 inhibitors were seen between LN-18 and U-87 MG cells. In LN-18 cells, a significant down-regulation of Nestin mRNA from 1.0 ± 0.21 (control) to 0.59 ± 0.12 (5 µM LY), 0.18 ± 0.09 (50 µM LY), 0.58 ± 0.17 (5 µM QT) and 0.22 ± 0.21 (50 µM TCS) was observed after treatment with PIM1 inhibitors. Such a decrease in Nestin mRNA was not caused in U-87 MG cells; instead, there was a significant up-regulation of Nestin, as already seen for CD133 at low-concentrated LY294002 (5 µM, 2.34 ± 1.30). SPARC was also differently modulated by PIM1 inhibitors in LN-18 cells compared to U-87 MG cells. As seen for Nestin, SPARC mRNA was significantly diminished from 1.0 ± 0.31 (control) to 0.47 ± 0.12 by 50 µM LY294002 and to 0.39 ± 0.19 by 50 µM TCS. A trend for a reduced SPARC mRNA level was also seen for 5 and 50 µM QT (0.70 ± 0.19 and 0.77 ± 0.32). In contrast, the treatment of U-87 MG cells did not cause a significant decrease of SPARC mRNA; instead, an elevated expression was again seen for 5 µM LY294002 (2.28 ± 1.54). GFAP mRNA was almost always under the limit of detection in LN-18 cells (cT >36 or undetermined or invalid), which is in accordance with published data [36], whereas U-87 MG cells expressed GFAP mRNA already at basal conditions. The treatment of U-87 MG cells with 5 and 50 µM LY294002 resulted in a significant about 4-fold (±2.23) and 10-fold (±5.71) up-regulation of GFAP mRNA. QT and TCS had no significant influence on GFAP mRNA content.

### 2.2. Impact of PIM1 Inhibitors on Stem Cell and Differentiation Markers in Human GBM Cells on Protein Level

Since mRNA levels do not always correlate well with the respective protein levels, we analyzed the protein amount of the differentiation markers (Figure 1C,D). The protein expression of CD44 was strongly enhanced by TCS in both LN-18 and U-87 MG GBM cells, from a relative value of 1.0 ± 0.02 (control LN-18) and 1.0 ± 0.04 (control U-87 MG) to 3.54 ± 2.08 (LN-18) and 4.26 ± 1.79 (U-87 MG) by 5 µM TCS, and to 2.81 ± 0.54 (LN-18) and 3.71 ± 1.80 (U-87 MG) by 50 µM TCS. A trend for an increased CD44 protein expression was also seen for LY294002 and QT. Regarding the CD133 protein, some differences in the response to PIM1 inhibitors were observed between LN-18 and U-87 MG GBM cells. In LN-18 cells, only treatment with 5 µM QT and 50 µM TCS resulted in a significantly decreased CD133 protein, from a relative value of 1.0 ± 0.12 (control) to 0.49 ± 0.19 (QT) and 0.38 ± 0.23 (TCS), respectively. In contrast, all tested PIM1 inhibitors provoked a significant reduction in CD133 protein in U-87 MG cells. For a concentration of 50 µM, a decrease of CD133 protein from a relative value of 1.0 ± 0.07 (control) to 0.53 ± 0.02 by LY294002, 0.22 (±0.08) by QT and 0.23 (±0.09) by TCS were observed. Some interesting findings were also obtained for Nestin protein. Whereas in U-87 MG GBM cells, an impact of PIM1 inhibitors was absent, LN-18 cells showed a strong down-regulation of Nestin protein after treatment with either LY294002, QT or TCS, particularly at a concentration of 50 µM, from a relative value of 1.0 (±0.06, control) to 0.56 (±0.22, LY), 0.40 (±0.28, QT) and 0.29 (±0.11, TCS), respectively. Regarding SPARC expression, the treatment of LN-18 cells with PIM1 inhibitors failed to cause a significant change in protein content, but a trend to an increased SPARC protein level was seen for 50 µM QT (3.74 ± 3.29) and 5 µM TCS (4.26 ± 2.45) in comparison to control cells. In U-87 MG cells, application of 50 µM TCS resulted in a significant increase of SPARC protein from a relative value of 1.0 ± 0.11 in control cells to 3.64 ± 1.07 upon TCS. LY294002 and QT only marginally influenced the SPARC protein content in U-87 MG cells. Determination of GFAP showed very consistent results in LN-18 and U-87 MG cells. In both cell lines, the astrocytic differentiation marker GFAP was nearly undetectable under basal conditions, but treatment with the PIM1 selective inhibitor TCS (50 µM) caused a significant and strong up-regulation of GFAP protein expression, to 253 ± 170 (LN-18) and 14.3 ± 11.4 (U-87 MG).

A similar decrease in CD133 and Nestin protein expression was seen when murine GL261 GBM cells were treated with 5 and 50 µM TCS, associated with an elevated protein level of GFAP (Appendix A).

### 2.3. Expression of PIM1 and Stem Cell Markers in Stem-like LN-18 Neurospheres

To analyze whether PIM1 expression is increased in GBM stem-like cells, we cultured LN-18 as so-called neurospheres (see Materials and Methods) which are described to be enriched in cancer stem cells [37,38]. A comparative presentation of adherent LN-18 cells and neurospheres is shown in Figure 2A. Our neurosphere experiments demonstrated that PIM1 mRNA was significantly up-regulated in LN-18 stem-like neurospheres, to about 4-fold in comparison to the adherent counterpart (Figure 2B). In contrast, PIM2 mRNA was unchanged, and PIM3 mRNA was tendencially decreased in LN-18 neurospheres but this effect was only significant for passage two. As already described before [39], mRNA expression of the well-accepted GBM stem cell markers Nestin and CD133 were also increased four to fivefold in stem-like neurospheres which served as a positive control for the stemness characteristics of LN-18 neurospheres. In addition, mRNA level of CD44, which is also discussed as potential GBM stem cell marker [39,40], was about fivefold higher in LN-18 neurospheres compared to adherent cells.

Regarding protein level, both the short (PIM1S, 35 kDa) and the long isoform (PIM1L, 45 kDa) of PIM1 were detectable (Figure 2C). The PIM1S protein level was not significantly different between adherent LN-18 cells and the respective stem-like neurospheres. In contrast, a significant (about sixfold) increase in PIM1L protein was found in LN-18 neurospheres in comparison to the adherent counterpart. As described in our previous work [39], we found an up-regulation of CD133 and Nestin protein in stem-like LN-18 neurospheres. For CD133, the elevated protein level was only significant for the second passage (about a 40-fold increase), since the variability between the three independent experiments for the third passage was very high. Regarding Nestin, LN-18 neurospheres showed a significantly increased protein content of the stem cell marker (3.7-fold for passage two and about 22-fold for passage three). Furthermore, CD44 protein was up-regulated in LN-18 neurospheres compared to adherent cells, but this was only significant for the second passage (about a fourfold increase).

### 2.4. Immunofluorescence Staining of PIM1 and Other Potential Stem Cell Markers in LN-18 Neurospheres

The results obtained by immunoblot analysis (Figure 2C) were verified by immunofluorescence staining. Figure 3 compares the immunofluorescence staining of PIM1 together with either CD44 (Figure 3A,B), CD133 (Figure 3C,D) or Nestin (Figure 3E,F) in adherent LN-18 cells (Figure 3A,C,E) and the respective stem-like neurospheres (Figure 3B,D,F) showing a strong PIM1 expression in LN18 neurospheres, whereas adherent cells expressed PIM1 only marginally (Figure 3B). Nearly the same enhanced protein expression in LN18 neurospheres was observed for CD44, CD133 and Nestin (Figure 3B,D,F) compared to the adherent cells (Figure 3A,C,E). Co-staining of PIM1 with the intermediate filament protein Nestin showed a partly overlapping localization of both proteins in LN-18 neurospheres, seen as yellow cellular structure (Figure 3F). In contrast, despite higher fluorescence signals for both CD133 and CD44 in LN-18 neurospheres compared to the adherent cells, an overlapping localization of these proteins with PIM1 was not clearly recognizable (Figure 3B,D).

### 2.5. Nestin and CD133 Expression after siRNA-Mediated Knockdown of PIM1

To confirm the results from pharmacological PIM1 blocking, the protein expression of the most accepted GBM stem cell markers Nestin and CD133 were investigated after siRNA-mediated knockdown of PIM1 in adherent LN-18 cells. As demonstrated in Figure 3G, both Nestin and CD133 were strongly reduced after 72 h, to 44.6% and 64%, respectively, in cells transfected with PIM1-specific siRNA compared to control transfected LN-18 cells. The efficiency of the siRNA-mediated PIM1 knockdown was checked on protein level. The protein expression of PIM1S was reduced to 33% and 26% for PIM1L, as indicated by Western blot analysis (Figure 3H). Similar results were obtained by immunofluorescence staining of PIM1 (Figure 3I). Of note, since PIM1 siRNA transfected cells showed very limited growth, it was necessary to pool the cells from three independent experiment to have enough protein for immunoblot analyses.

Recently, we have shown that specific inhibition of PIM1 by TCS PIM1-1 (TCS) causes viability loss, cell cycle arrest and apoptosis in different GBM cell lines (LN-18, U-87 MG and GL261) [23]. To investigate whether inhibition of PIM1 also leads to cytotoxic effects in stem-like LN-18 neurospheres, we treated LN-18 neurospheres with TCS (specific PIM1 inhibitor) [35] and LY294002 (dual PI3K/PIM1 inhibitor) [33] and compared the effects with temozolomide as the standard therapeutic agent in GBM. In Figure 4F, the viability of stem-like LN-18 neurospheres after treatment for 7 days is shown. Whereas temozolomide (100 µM) showed no influence on the viability of LN-18 neurospheres, TCS strongly impaired the cell viability to 8.86 ± 11.4% (20 µM) and 2.56 ± 3.25% (50 µM). Furthermore, treatment with the dual PI3K/PIM1 inhibitor LY294002 (50 µM) also resulted in a significantly decreased viability of LN-18 neurospheres (3.87 ± 3.93%), to the same extent as TCS. For the used concentrations of TCS and LY294002 (50 µM), we recently observed inhibitory effects on PIM1 kinase function by measuring the phosphorylation status of the PIM1 substrate BAD [23].

### 2.6. Neurosphere Formation after PIM1 Blocking

For further validation of the cytotoxic influence of PIM1 inhibition, we evaluated the neurosphere formation of LN-18 cells by microscopic determination of the number and size of neurospheres after the blocking of PIM1. Figure 4A,B shows representative microscopic images of LN-18 neurospheres, three and seven days after application of either the solvent DMSO, 50 µM TCS, 50 µM LY294002 (LY) or 100 µM temozolomide (TMZ). In particular, an almost complete loss of neurosphere formation was observed seven days after the application of TCS or LY294002. In contrast, when treated with 100 µM TMZ, we observed that although many neurospheres were formed, they were smaller in size. The statistical evaluation of the number and sizes of the LN-18 neurospheres is demonstrated in Figure 4C,D. It is clearly visible that specific blocking of PIM1 resulted in a strongly impaired neurosphere formation, both after three and seven days, which was more pronounced compared to dual PI3K/PIM1 inhibition by LY294002. On day three, the formation of small neurospheres (40–100 µm) and neurospheres of medium size (101–150 µm) were reduced to half and one third, respectively, by 50 µM TCS compared to the control values (DMSO, solvent). Large (151–200 µm) and very large (>200 µm) neurospheres were virtually unavailable after three days of TCS treatment. This cytotoxic effect of the PIM1 inhibitor TCS was even stronger on day seven, and intact LN-18 neurospheres were nearly absent (Figure 4D). The dual PI3K/PIM1 antagonist LY294002 showed inhibitory effects on LN-18 neurosphere formation as early as three days and especially seven days after application, with significantly smaller neurospheres compared to the control. Interestingly, the standard chemotherapeutic TMZ had much lower effects on neurosphere formation in comparison to TCS and LY294002.

In Figure 4E, the effect of 1, 12.5, 25 and 50 µM TCS on the LN-18 formation of neurospheres is illustrated as a graph. The application of 1 µM TCS showed no significant effects on the growth of LN-18 neurospheres, but a concentration of 12.5 µM TCS was sufficient to strongly disturb both small and large neurospheres (Figure 4F).

### 2.7. PIM1 Blocking Reduced DNA Synthesis Rate and Induced Apoptosis

The loss of LN-18 neurosphere viability and formation was accompanied by a reduced DNA synthesis rate as determined by the BrdU assay. Data shown in Figure 5A,B demonstrate that the DNA synthesis rates of both LN-18 adherent and neurosphere cells were significantly reduced after treatment with the PIM1 blocker TCS and LY294002 for 72 h. In adherent LN-18 cells, a TCS concentration of 12.5 µM was already enough to significantly reduce the DNA synthesis rate to 36% (±18.4), whereas this was not the case in LN-18 neurospheres. Nonetheless, treatment with 25 and 50 µM TCS resulted in a similarly decreased DNA synthesis rate, to below 5%. Interestingly, the effect of the dual PI3K/PIM1 inhibitor LY294002 on DNA synthesis was somewhat stronger in LN-18 neurospheres (23.8 ± 6.3%), compared to the adherent counterpart (42 ± 18.3%). The standard chemotherapeutic temozolomide showed no influence on DNA synthesis in both adherent and neurosphere cells (Figure 5A,B).

Furthermore, caspase 3 activation was investigated to test whether apoptosis is induced by PIM1 blocking. In contrast to TMZ, which had no effect on caspase 3 activity, application of both TCS and LY294002 resulted in a partially significant increase of caspase 3 activity in LN-18 adherent cells (Figure 5C) and neurospheres (Figure 5D), depending on the concentration used. However, direct comparison showed that the induction of caspase 3 by TCS was stronger in LN-18 neurospheres, especially at a concentration of 25 µM, with an activity value of 260 ± 67.8% in adherent cells and 620 ± 417.4% in neurospheres. On the other hand, as seen for the DNA synthesis rate, LY294002 showed more elevated caspase 3 activity in adherent LN-18 cells (422.8 ± 231.9%) compared to LN-18 neurospheres (210 ± 121.8%). Caspase 3 activity was also significantly increased, by 50 µM TCS PIM1-1 and LY294002, to 189 ± 41.4% and 264 ± 69.9% in adherent U-87 MG cells, which was somewhat lower in U-87 MG neurospheres, with values of 159 ± 35.1% and 237 ± 65.3%, respectively (Figure 5E,F).

As a proliferation marker, we additionally analyzed the expression of PCNA (Proliferating Cell Nuclear Antigen) in U-87 MG cells. As seen in Figure 5G,H, PCNA protein expression was significantly reduced by TCS PIM1-1 in both adherent and stem-like U-87 MG neurospheres, to a similar extent, with values of 35.8 ± 24.2% (adherent cells) and 35.8 ± 18.3% (neurospheres), by treatment with 50 µM TCS for 72 h.

## 3. Discussion

The functional relationship between PIM1 and cancer has been well established in recent years [41], but there is limited knowledge of its precise role in glioblastoma (GBM) and, in particular, GBM stem cell behavior. In previous studies, our group has shown the preclinical pharmacological efficacy of the PIM1 antagonist TCS PIM1-1 that inhibits GBM growth in an orthotopic mouse model [23]. In this study, we show, for the first time, that PIM1 is associated with stem cell properties of GBM cells. To study PIM1 function in GBM stem cell behavior, we used the well-accepted human GBM cell lines LN-18 and U-87 MG. Both LN-18 and U-87 MG have been shown to exhibit brain tumor stem cell capacities and are capable of forming stem-like neurospheres [42,43]. We incubated GBM cells with different PIM1 antagonists to compare the individual effects of the compounds and thereby characterize what might be common to all three inhibitors. LY294002 has been described as a combined PIM1 and PI3K inhibitor [33], but also has other independent effects such as inhibition of the protein kinase CK2 [44] or GSK3a and b [45]. In addition, the selective PIM1 inhibitor quercetagetin (QT) [34] and the specific PIM1 inhibitor TCS PIM1-1 (TCS) [35] were used. In direct comparison, LN-18 and U-87 MG cells responded more strongly to the specific PIM1 inhibition by QT and especially TCS than to the unspecific LY294002 inhibitor, which may be due to the opposing effects of blocking more than one target protein.

To evaluate the impact of PIM1 inhibition on GBM stem cell features, we examined the expression of CD133, since several studies have demonstrated a close correlation between the expression of the stem cell marker CD133, chemoresistance and GBM survival [37,46,47]. All of the investigated PIM1 inhibitors decreased CD133 protein in both LN-18 and U-87 MG cells, but not CD133 mRNA. This presents an argument for post-transcriptional or (post)translational regulatory effects, possibly based on a reduced phosphorylation of proteins involved in the de/stabilization of CD133. Besides the possibility of reduced CD133 protein stability after PIM1 inhibition, a diminished translation rate of mRNA to protein is also imaginable. As in our study, Sgambato and colleagues found no relationship between the CD133 protein and mRNA [48]. It is noteworthy that ubiquitin-specific proteases (USPs), known to stabilize proteins such as USP22, have been shown to be involved in the regulation of CD133 [49,50]. To our knowledge, a regulation of USPs by PIM1 has not been shown to date but, in addition to other ways of protein stabilization, could be an explanation for the observed increase in the CD133 protein.

CD133 was shown to be co-expressed with CD44, the receptor for the glycosaminoglycan hyaluronate [51], in GBM neurospheres [52], and inhibition of CD44 affects GBM progression in mice [53]. In accordance therewith, an increase in the CD44 protein was similarly found in both LN-18 and U-87 MG stem-like cells. Unexpectedly, PIM1 inhibition also caused an increased CD44 protein content, particularly by the specific PIM1 blocker TCS PIM1-1. The molecular background of this is unclear, as a direct regulation of CD44 by PIM1 has not been described so far. Further, it cannot be excluded that PIM1 inhibition eliminates CD133 and Nestin positive stem-like cells, but not the CD44 positive ones, resulting in the observed enhanced expression of CD44.

Nestin is also discussed as playing a role in the cell survival and proliferation of cancer stem cells [54], and has been shown to be a strong prognostic factor for glioma malignancy [55]. While a significant down-regulation of Nestin was observed in LN-18 cells, U-87 MG cells did not show such a modulation after application of PIM1 inhibitors. Interestingly, p53 has been shown to limit the expression of Nestin, and the loss of wild-type p53 facilitates dedifferentiation of mature hepatocytes into Nestin-positive progenitor-like cells [56]. In our study, LN-18 cells, which have a mutant p53 status [57,58], showed higher Nestin protein expression than U-87 MG cells with wild-type p53. Further, LN-18 cells carry wild-type PTEN phosphatase, whereas U-87 MG cells are negative for PTEN; a direct influence of PTEN on Nestin expression or a regulation of PTEN by PIM1 is—to our knowledge—not described. Of note, activation of AKT (protein kinase B), which is negatively regulated by PTEN [59], correlates with Nestin expression in the perivascular niche [60], the location of residing glioma stem cells. Therefore, it may be possible that PTEN is also involved in the down-regulation of Nestin by PIM1 inhibitors.

In addition, we found that expression of the astrocytic marker GFAP, which is also expressed in glioma cells [54,61], is different between LN-18 and U-87 MG cells. GFAP mRNA was below the detection limit in LN-18 cells, whereas U-87 MG cells showed a slight GFAP mRNA expression even under basal conditions, which has also been described by others [62]. However, in terms of protein level, we could not detect GFAP in neither LN-18 not U-87 MG cells under basal conditions. Interestingly, specific blocking of PIM1 by TCS PIM1-1 resulted in a strong up-regulation of the GFAP protein in both cell lines. The nuclear expression of mutant p53 and cytoplasmic expression of GFAP are known to be inversely correlated [63], and transfection of glioma cells with wildtype p53 results in overexpression of GFAP [64]. Of note, PIM1 destabilization activates p53-dependent signaling pathways in cancer cells [65]. Therefore, it might be possible that GFAP re-expression observed after application of the specific PIM1 inhibitor TCS PIM1-1 is partially dependent on p53, or other coupled signaling cascades.

Another marker for astrocytes and glial cells is the Secreted protein, acidic, and rich in cysteine (SPARC), which is also highly expressed in glioma [66,67]. In terms of protein level, SPARC was induced by all PIM1 inhibitors, in both LN-18 and U-87 MG cells, but this increase was only significant for the specific inhibitor TCS PIM1-1 in U-87 MG cells. The modulation of SPARC by PIM1 kinase has not been previously reported in the literature, and thus the molecular background of SPARC regulation after PIM1 inhibition remains unclear. Of note, an association between increased SPARC expression and decreased tumor proliferation was found, but on the other hand, SPARC was shown to induce migration of GBM cells [67,68]. Therefore, up-regulation of SPARC by PIM1 inhibition could be a double-edged sword: it could decrease proliferation but increase GBM cell invasion. However, our own studies using the PIM1 inhibitor TCS PIM1-1 did not show any influence of PIM1 inhibition on GBM cell migration [23].

However, knockdown of PIM1 by siRNA revealed a strong reduction of the two most widely accepted glioma stem cell markers, CD133 and Nestin. Since PIM1 siRNA causes knockdown of both PIM1S and PIM1L, it is not possible to distinguish between PIM1 isoform-specific effects on stem cell marker expression. Nevertheless, our observation that LN-18 neurospheres have increased PIM1 mRNA and protein levels of the long isoform PIM1L suggests PIM1 may have a role in regulating the GSCs phenotype. In contrast, the expressions of PIM1S and PIM2 were not significantly altered, and PIM3 was even down-regulated. These results provide an argument for PIM1L having a more relevant function in GSCs behavior. In prostate cancer, PIM1S and PIM1L are up-regulated and play a key role in maintaining androgen receptor (AR) stability and transcriptional activity. While PIM1L is capable of AR phosphorylation at Thr-850, leading to AR stabilization, PIM1S can promote the degradation of AR through phosphorylation at Ser-213 [69]. Such a distinct function of PIM1S and PIM1L may also be plausible in GBM cells, and the observed up-regulation of PIM1L could stabilize proteins important for stem cell behavior.

Using TCS PIM1-1, we recently showed the reduced growth of adherent GBM cells [23]. In this study, we investigated whether PIM1 antagonism is also effective against stem-like neurospheres. Cell viability of LN-18 GBM neurosphere cells was strongly reduced by both LY294002 as a dual PI3K/PIM1 inhibitor and the selective PIM1 antagonist TCS PIM1-1, whereas temozolomide was ineffective. The cytotoxic effect of PIM1 blocking was accompanied by strongly reduced DNA synthesis, a diminished expression of the proliferation marker PCNA and the induction of apoptosis. The growth-inhibitory effect of PIM1 blockade was confirmed by microscopic imaging and counting neurospheres. After treatment with LY294002 and TCS PIM1-1, both the size and number of neurospheres were significantly reduced. Thus, PIM1 inhibition is able to kill both normal GBM cells [23] and stem-like neurosphere cells. While an effect on the size and cell viability of neural stem cell neurospheres has been shown for LY294002 [70,71], to our knowledge, this is the first report showing such a repressive effect by TCS PIM1-1. Whether targeting PIM1 alone, or in combination with other potential regulators of GSCs, is sufficient to stop tumor growth in GBM patients remains to be elucidated in future studies.

## 4. Conclusions and Study Limitations

Besides the described general relevance of PIM1 in cancer, our study shows, for the first time, that the inhibition of PIM1 affects glioblastoma stem cell behavior and survival, suggesting PIM1 targeting may be a potential anti-glioblastoma therapy. However, our study also has limitations. We used two well-recognized cell lines (LN-18 and U-87 MG) to analyze the role of PIM1. High genetic variation and rapid genetic diversification have been demonstrated for different cell lines, which may alter cancer cell line phenotype and drug response [72]. Thus, experiments with primary GBM cells freshly isolated from tumor tissue could complement the results of our study. Of note, the neurosphere assay used in our experiments is considered the gold standard for identifying the stem cell population in brain tissue. Neurosphere culturing is more relevant for modelling brain tumor biology than traditional 2D culture conditions, but important drawbacks of neurosphere assays are their inapplicability for detection of quiescent CSCs and the lack of the in vivo tumor environment [73,74,75,76]. To address the importance of the tumor microenvironment for GSCs, future studies should include in vivo animal models with inoculation of PIM1 knockout GBM (neurosphere) cells, for example, generated by the CRISPR/Cas9 technology, to finally assess the impact of PIM1 on GBM pathogenesis.

## 5. Materials and Methods

### 5.1. Cultivation of GBM Adherent Cells and Neurospheres

The maintenance of adherent LN-18 cells and U-87 MG cells (both human, authenticated and characterized by ATCC, Manassas, VA, USA) as well as of murine GL261 cells (kindly provided by Prof. Michael Synowitz, University Kiel) was performed in DMEM medium supplemented with 10% FCS, 2 mM glutamine and 1% NEAA solution (100×) (all from PAN Laboratories, Cölbe, Germany) at 37 °C, 95% humidity and 5% CO_2_. Neurospheres, which are thought to be enriched in cancer stem cells [30,31], were cultured using the NeuroCult NS-A Proliferation Kit (STEMCELL^TM^ Technologies, Cologne, Germany) according to the manufacturers’ protocol.

For inhibition experiments, the following PIM1 inhibitors were used: LY294002, quercetagetin, (both Calbiochem, Bad Soden, Germany) and TCS PIM1-1 (TOCRIS biosciences, Wiesbaden, Germany). In some experiments, cells were treated with temozolomide (Sigma Aldrich, Munich, Germany). The used concentrations of the compounds are listed in the respective figures.

### 5.2. Real-Time PCR Analysis

Total RNA was isolated using PeqGold RNAPure (PeqLab, Erlangen, Germany) and reversely transcribed using the High Capacity cDNA Reverse Transcription Kit (Applied Biosystems™, Thermo Scientific, Schwerte) according to the manufacturers’ protocol. The following Gene Expression Assays on Demand from Applied Biosystems were used: CD44, Hs01075861_m1; CD133, Hs01009250_m1; GFAP, Hs00909236_m1; Nestin, Hs04187831_g1; PIM1, Hs01065498_m1; PIM2, Hs00179139_m1; PIM3, Hs00420511_g1; Sparc, Hs00234160_m1; endogenous control eukaryotic 18S rRNA, Hs03003631_g1; TBP, Hs00427620_m1; GAPDH, Hs02758991_g1; ß-Actin, 4310881E-0910026. Quantitative Real-Time PCR was performed on a 7900 HT Fast Real-Time PCR system (Applied Biosystems™, Thermo Scientific, Schwerte) and QuantStudio 12 Flex Real-Time PCR System (life technologies, Applied Biosystems™, Darmstadt). mRNA content of each target gene was normalized to 18S rRNA (Figure 1) or the mean of 18S rRNA, TBP, β-Actin and GAPDH (Figure 2), and is expressed relative to control cells.

### 5.3. Western Blot

For preparation of protein lysates, harvested cells were incubated on ice for 30 min with the following lysis buffer: 50 mM Tris-HCl pH 7.4, 100 mM NaCl, 0.1% Triton X-100 and 5 mM EDTA containing protease/phosphatase inhibitors (1 mM leupeptin, 1 mM aprotinin and 250 μg/mL sodium vanadate). The BCA Protein Assay Kit (Thermo Fisher Scientific, Rockford, IL, USA) was used to determine the protein concentration of cell lysates. Subsequently, after denaturation in Laemmli SDS sample buffer at 95 °C for 5 min, 30 to 40 µg of each sample were separated on SDS polyacrylamide gels. The tank blot system (Bio-Rad, Hempstead, UK) was used for immunoblotting of separated proteins onto a nitrocellulose membrane (GE Healthcare Life Science, Buckinghamshire, UK). After blotting, membranes were blocked in 5% FCS or skim milk in Tris-buffered saline containing 0.05% Tween 20 (TBST) and 1% BSA for 1 h at room temperature. Used antibodies as well as dilution and incubation conditions are described in the Appendix A.

### 5.4. Immunofluorescence Microscopy

To determine the cellular localization of PIM1, CD44, CD133 and Nestin in adherent LN-18 cells compared to neurospheres, we performed immunofluorescence microscopy using the Zeiss LSM780 system (Carl Zeiss, Jena, Germany). Cells were grown for 7 days in stem cell media (neurospheres) or DMEM medium (adherent monolayer cells) as described above. Neurospheres were harvested at 500× *g* at room temperature for 3 min. After three washing steps with phosphate buffered saline (PBS), cells were fixed in ethanol (99.9%, Chemsolute, Th. Geyer GmbH & Co. KG, Renningen, Germany) containing 5% acetic acid for 10 min at −20 °C. Fixation was removed by centrifugation at 500× *g* followed by three washing steps with PBS. Afterwards, cell pellets were resuspended in one drop of Jung Tissue Freezing Medium (Leica Microsystems Vertrieb GmbH, Wetzlar, Germany) and 1 µL of a 0.05% trypan blue for visualization of the cells during cryosectioning [32]. Cells were frozen at −20 °C in Tissue-Tek Cryomolds (Sakura Finetek USA, Inc. Torrance, CA, USA) with Jung Tissue Freezing medium and stored at −20 °C. Ultrathin sections (7 µm) were sliced with a Feather microtome blade (Type A35) on a Leica CM1900 cryostat, transferred to microscope slides, surrounded by a Liquid Blocker Super Pap Pen (Daido Sangyo Co., Ltd. Tokyo, Japan), and stored at −20 °C until use. The staining procedure is described in the Appendix A.

### 5.5. Cell Viability Analysis

Neurosphere cells were seeded at 200,000 cells/well in 6-well multiplates in a medium containing one of the following inhibitors: LY294002, quercetagetin, TCS PIM1-1 or temozolomide (Sigma Aldrich, Munich, Germany). After the respective incubation period, the medium was replaced with fresh medium containing 10% resazurine (PromoCell, Heidelberg, Germany), and cells were incubated for two hours at 37 °C. Fluorescence was determined on a microplate reader (excitation, 530 nm; emission, 590 nm) (TECAN infinite M200, Tecan GmbH, Crailsheim, Germany).

### 5.6. BrdU Incorporation Assay for Determination of DNA Synthesis

For BrdU incorporation assay (Cell Proliferation ELISA, BrdU, Roche Diagnostics GmbH, Germany), LN-18 cells or U-87 MG cells were seeded in 96-well plates and incubated for four days either in FCS containing DMEM medium for adherent cells, or in NeuroCult™ NS-A Proliferation medium to get neurospheres. Then, treatment with PIM1 inhibitors or temozolomide was started for another three days. Afterwards, BrdU incorporation assay was performed according to the manufacturers’ protocol.

### 5.7. Caspase 3 Activity Assay

Caspase 3 activity assay was performed using a commercially available kit (BioVision Inc, Milpitas, CA, USA). LN-18 cells or U-87 MG cells were seeded in 6-well plates and cultured as adherent cells or neurospheres for three days, followed by incubation with the respective compounds for 48 h. After treatment, the supernatants, including cells detached due to cell death, were collected in a conical tube. Adherent cells were scraped, collected in the same tube as the supernatant and centrifuged at 250× *g* for 10 min. Afterwards, cells were prepared according to the manufacturer’s protocol as described in Appendix A.

### 5.8. Measurement of Neurosphere Number and Sizes

For determination of the impact of PIM1 inhibitors on neurosphere formation, we analyzed both the number and sizes of neurospheres after treatment. LN-18 neurospheres were seeded at 200,000 cells/well in 6-well multiplates and treated with the compounds as described in the respective figures. Randomization was ensured by taking pictures at defined areas of the 6-well plates at day 3 and day 7, using an automated Axio Observer Z1 microscope, followed by analyses of the number and sizes of neurospheres using the Axio Vision Rel. 4.9.1.0 software.

### 5.9. Small Interfering (si)RNA-Mediated Knockdown of PIM1

Lipofectamine 2000 reagent (Invitrogen by ThermoFisher Scientific, Darmstadt, Germany) was used for transfection of LN-18 cells with siRNA according to the manufacturer’s instructions one day after seeding of cells in 12-well plates at a density of 200,000 cells/well. PIM1 siRNA and control siRNA (OriGene Technologies, Rockville, MD, USA) were used at a final concentration of 5 nmol. To optimize the knockdown efficiency, the transfection was repeated after 24 h. Using immunoblot analysis, the knockdown efficiency of PIM1 was tested 72 h post first transfection (Figure 4C, right panel) as described below.

### 5.10. Statistical Analysis

Graph Pad Prism 5.0 (GraphPad Prism Software, San Diego, CA, USA) was used for data presentation and statistical calculations. Analyses represent the results of three to five independent experiments, as indicated in the respective figure legends, and are shown as means plus standard deviation (SD). The Student’s t-test and the Mann–Whitney U test were used for comparisons between two groups. More than two groups were compared by OneWay ANOVA and Dunnett’s multiple comparison test. Statistical significance was defined as *p*-value < 0.05 (* < 0.05, ** < 0.01 and *** < 0.001).

## Figures and Tables

**Figure 1 ijms-22-11126-f001:**
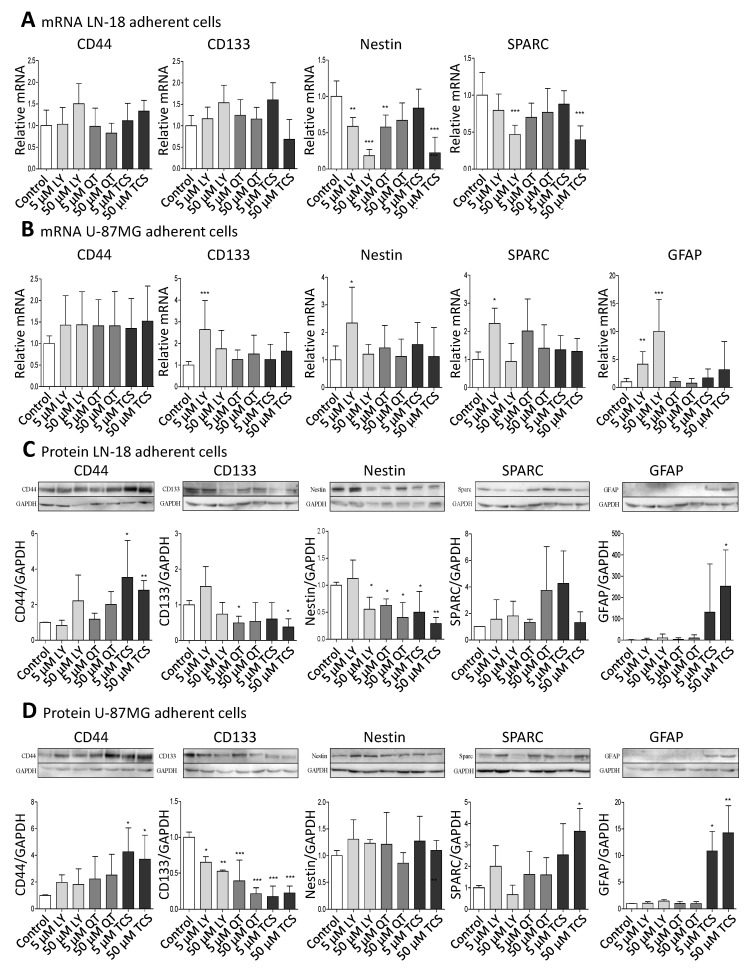
Pharmacological PIM1 blocking regulates expression of stemness and differentiation marker proteins in GBM cells. Human GBM cells (LN-18, U-87 MG) were treated each with 5 or 50 µM of the dual PI3K/PIM1 inhibitor LY294002 (LY), or the selective PIM1 inhibitors quercetagetin (QT) and TCS PIM1-1 (TCS). The respective solvent, dimethylsulfoxide (DMSO), was added to the control cells. (**A**,**B**), mRNA level of CD44, CD133, Nestin, SPARC and GFAP in LN-18 (**A**) and U-87 MG (**B**) cells determined by quantitative RT–PCR. mRNA content of each target gene was normalized to 18S rRNA and is expressed relative to control GBM cells. Of note, in LN-18 GBM cells, mRNA expression of GFAP was under the limit of detection (cT >36 until undetermined). Results are representative of three independent experiments. Columns represent means and SD, significant differences for * *p* < 0.05, ** *p* < 0.005 and *** *p* < 0.001. (**C**,**D**), Protein expression of CD44, CD133, Nestin, SPARC and GFAP in LN-18 (**C**) and U-87 MG (**D**) cell lysates by immunoblotting using specific antibodies. GAPDH served as loading control. Protein band intensities were quantified by densitometric analysis (Quantity One, Bio-Rad). The relative optical densities of the specific bands were calculated and normalized to GAPDH. Columns represent means and SD of four or five independent experiments, significant differences for * *p* < 0.05, ** *p* < 0.005 and *** *p* < 0.001.

**Figure 2 ijms-22-11126-f002:**
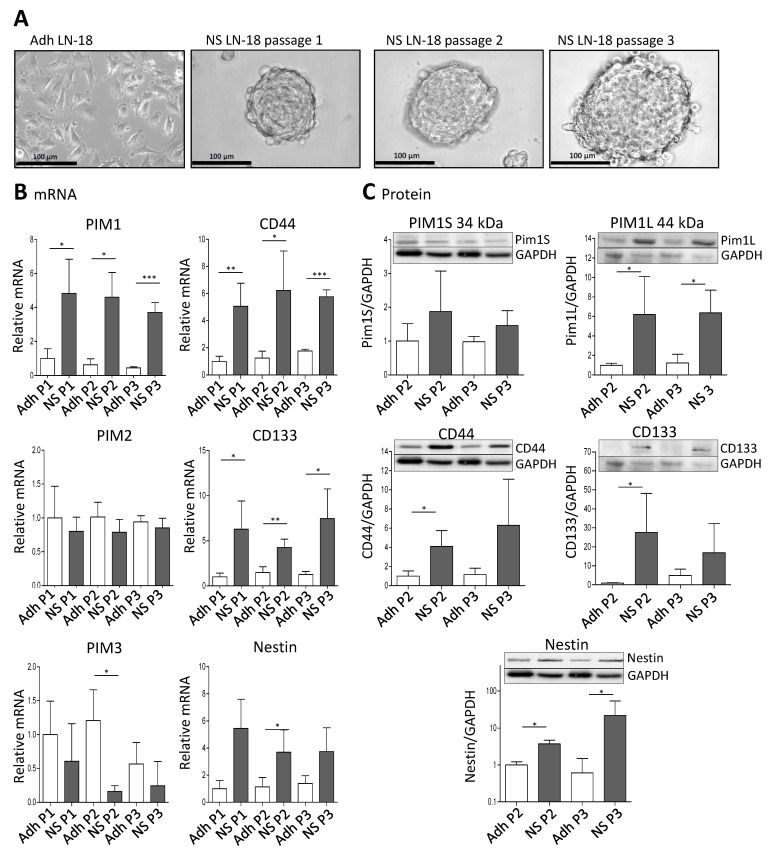
Expression of PIM1 and stem cell marker in LN-18 neurospheres is elevated. Human LN-18 GBM cells were cultured either as adherent monolayer cells or under specific conditions as three-dimensional neurospheres. (**A**) Representative morphological images of LN-18 adherent monolayer cells (Adh) and neurospheres (NS) at passages one, two and three using phase contrast microscopy. Scale bars represent 100 µm. (**B**) mRNA levels of PIM kinases (PIM1, PIM2 and PIM3, left panel) as well as of the stem cell markers CD44, CD133 and Nestin, determined by quantitative RT–PCR. mRNA content of each target gene was normalized to the mean of 18S rRNA, TBP, GAPDH and β-Actin gene expression, and is shown relative to adherent LN-18 cells. Results are representative of three (passages one and two) or four (passage 3) independent experiments. Columns represent means and SD, significant differences for * *p* < 0.05, ** *p* < 0.005 and *** *p* < 0.001. (**C**) Protein expression of both PIM1 kinase isoforms (PIM1S and PIM1L) as well as CD44, CD133 and Nestin in LN-18 adherent and neurosphere cell lysates (passages two and three) by immunoblot analysis using specific antibodies. GAPDH served as a control. Results are representative of three or four independent experiments. Protein band intensities were quantified by densitometric analysis (Quantity One, Bio-Rad). The relative optical densities of the specific bands were calculated and normalized to GAPDH. Columns represent means and SD; significant differences for * *p* < 0.05.

**Figure 3 ijms-22-11126-f003:**
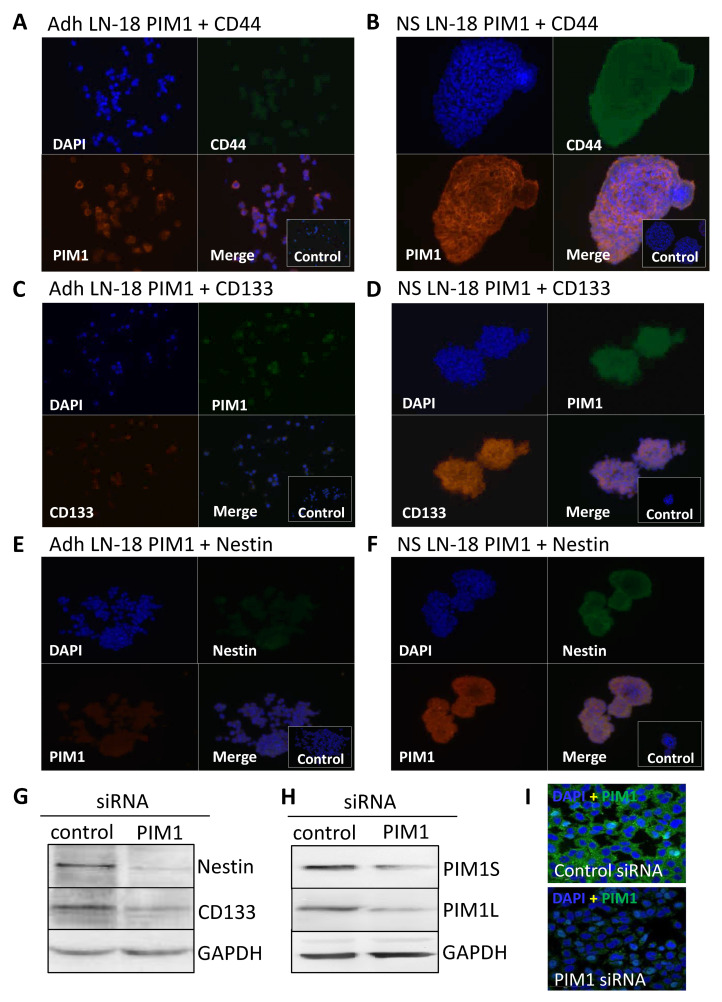
Co-Expression of PIM1 and stem cell markers in GBM neurospheres and after siRNA-mediated knockdown of PIM1. (**A**–**F**) Dual immunofluorescence staining of PIM1 (Alexa Fluor^®^ 568) with either CD44 (**A**,**B**) or with Nestin (**E**,**F**) (both Alexa Fluor^®^ 488) and PIM1 (Alexa Fluor^®^ 488) with CD133 (**C**,**D**) (Alexa Fluor^®^ 568) in adherent LN-18 monolayer cells (**A**,**C**,**E**) and the respective neurospheres (**B**,**C**,**F**). The cell nuclei were counterstained using DAPI (blue). Representative images assessed via immunofluorescence microscopy. Cells were also stained with the Alexa Fluor 488^®^/Alexa Fluor 568^®^ secondary antibodies alone as a control. (**G**) Expression of Nestin and CD133 after knockdown of PIM1 by siRNA determined by immunoblotting. A representative blot is shown; a non-targeting, scrambled shRNA served as control. (**H**,**I**) Confirmation of siRNA-mediated knockdown of PIM1S and PIM1L. (**H**) A representative Western blot is shown; detection of GAPDH served as loading control. (**I**) Immunofluorescence staining of PIM1 in LN-18 cells after transfection with control siRNA or PIM1 specific siRNA confirming down-regulation of PIM1. The cell nuclei were counterstained using DAPI (blue). Representative images assessed via immunofluorescence microscopy. Objective of the Zeiss LSM 780 laser scanning confocal microscope: 40x/1.4 DIC (oil).

**Figure 4 ijms-22-11126-f004:**
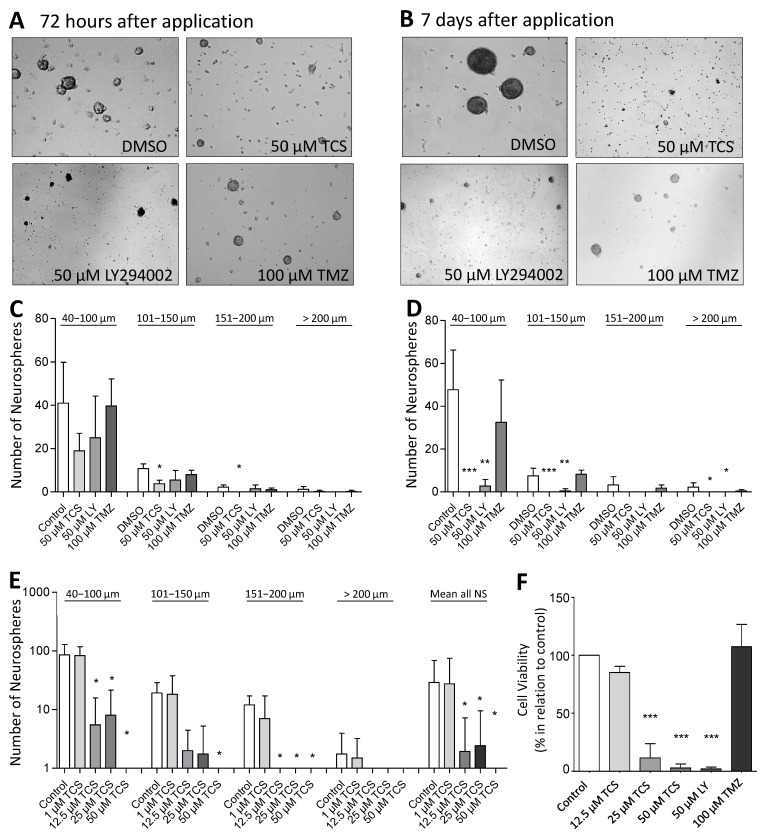
Pharmacological inhibition of PIM1 promotes loss of GBM cell viability and impairs formation of neurospheres. (**A**,**B**) PIM1 inhibition significantly impairs neurosphere formation. LN-18 cells were grown as neurospheres for 7 days before treatment with either the selective PIM1 inhibitor TCS PIM1-1 (TCS; 50 µM), the dual PI3K/PIM1 inhibitor LY294002 (50 µM) or temozolomide (TMZ; 100 µM), started for 72 h (**A**) or 7 days (**B**). Control cells were treated with the respective solvent DMSO. Representative images of neurospheres at the defined time points are shown. (**C**,**D**) Statistical analysis of the average sphere size after treatment with either the selective PIM1 inhibitor TCS PIM1-1 (TCS, 50 µM), the dual PI3K/PIM1 inhibitor LY294002 (LY; 50 µM) or temozolomide (TMZ; 100 µM) for 72 h (**C**) or 7 days (**D**). Neurospheres were subdivided according to their sizes in four groups (40–100 µm, 101–150 µm, 151–200 µm and >200 µm), which were separately investigated. Control cells were treated with the solvent DMSO. Columns represent means and SD of four independent experiments, significant differences for * *p* < 0.05, ** *p* < 0.005 and *** *p* < 0.001. (**E**) Growth characteristics of LN-18 neurospheres after application of increasing TSC PIM1-1 concentrations (TCS; 1, 12.5, 25 and 50 µM). Statistical analysis of the differently sized neurospheres (40–100 µm, 101–150 µm, 151–200 µm and >200 µm) and the total neurosphere number after TCS treatment are shown. Columns represent means and SD of four independent experiments, significant differences for * *p* < 0.05. (**F**) Cell viability of LN-18 neurospheres determined by the Resazurine Assay. LN-18 cells were grown as neurospheres for 7 days before treatment with either the selective PIM1 inhibitor TCS PIM1-1 (TCS; 12.5, 25 and 50 µM), the dual PI3K/PIM1 inhibitor LY294002 (LY; 50 µM), temozolomide (TMZ; 100 µM) or the solvent DMSO (control) was started for further 7 days. Columns represent means and SD of four independent experiments, significant differences for *** *p* < 0.001.

**Figure 5 ijms-22-11126-f005:**
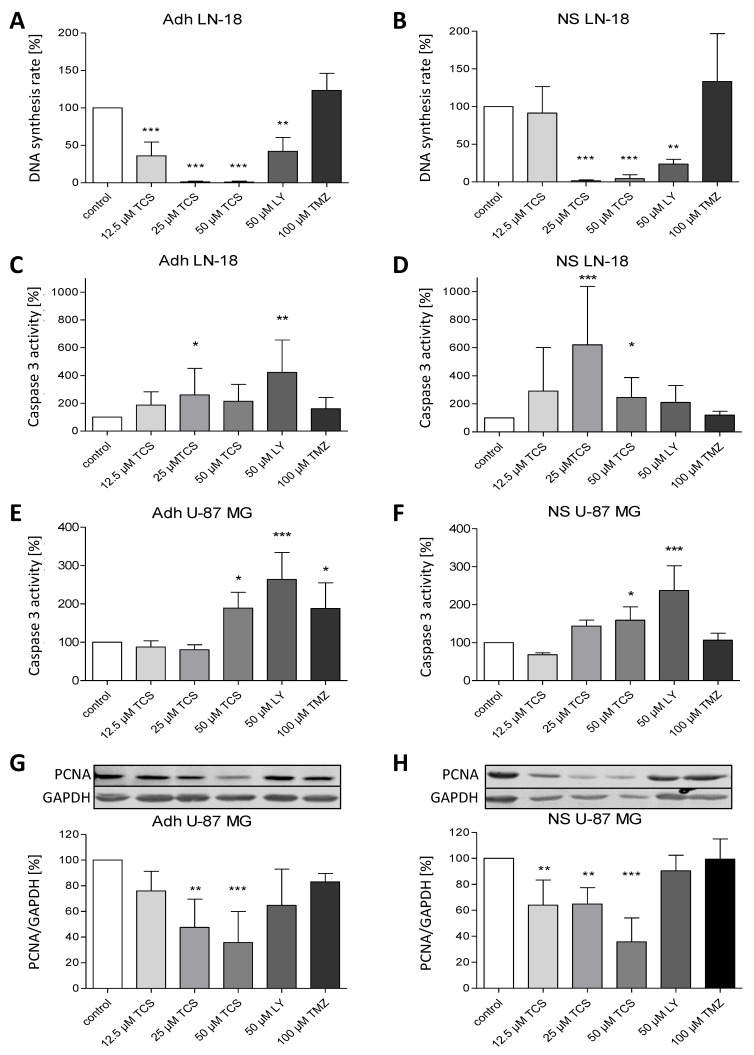
Pharmacological inhibition of PIM1 promotes a reduction in DNA synthesis rate and apoptosis. (**A**,**B**) BrdU cell proliferation assay was performed on LN-18 adherent cells (**A**) and neurospheres (**B**) to determine the DNA synthesis rate after PIM1 inhibition with TCS PIM1-1 (TCS; 12.5, 25 and 50 µM), LY294002 (LY; 50 µM) and Quercetagetin (QT; 50 µM) as well as treatment with 100 µM temozolomide (TMZ) for 72 h. Columns represent means and SD (*n* = 4), significant differences for ** *p* < 0.005 and *** *p* < 0.001. (**C**,**D**) Effects of TCS PIM1-1 (TCS; 12.5, 25 and 50 µM), LY294002 (LY; 50 µM) and Quercetagetin (QT; 50 µM) as well as 100 µM temozolomide (TMZ) on caspase 3 activity after 72 h in LN-18 adherent cells (**C**) and neurospheres (**D**) analyzed by a commercially available Caspase 3 fluorometric assay. Results are presented as the means and SD (*n* = 8–9). * *p* < 0.05, ** *p* < 0.005 and *** *p* < 0.001 in comparison to DMSO (control). (**E**,**F**) Effects of TCS PIM1-1 (TCS; 12.5, 25 and 50 µM), LY294002 (LY; 50 µM) and Quercetagetin (QT; 50 µM) as well as 100 µM temozolomide (TMZ) on caspase 3 activity after 72 h in U-87 MG adherent cells (**E**) and neurospheres (**F**) analyzed by a commercially available Caspase 3 fluorometric assay. Results are presented as the means and SD (*n* = 4–5), * *p* < 0.05, and *** *p* < 0.001 in comparison to DMSO (control). (**G**,**H**) Expression of PCNA as proliferation marker determined by immunoblot analyses after treatment of U-87 MG adherent cells (**G**) and neurospheres (**H**) with TCS PIM1-1 (TCS; 12.5, 25 and 50 µM), LY294002 (LY; 50 µM) and Quercetagetin (QT; 50 µM) as well as 100 µM temozolomide (TMZ) for 72 h. GAPDH was used as loading control for normalization. Results are presented as the means and SD (*n* = 4–5). ** *p* < 0.005 and *** *p* < 0.001 in comparison to DMSO (control).

## Data Availability

Not applicable.

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
