# Peer review of "PIM1 Inhibition Affects Glioblastoma Stem Cell Behavior and Kills Glioblastoma Stem-like Cells"

_ijms, 2021, doi:10.3390/ijms222011126_

Round 1

Reviewer 1 Report

Dear authors ...

I read this work with interest .... overall it seems to me well done and very clear with meaningful experiments and good figures and results. However, I suggest some changes to make it even more effective.

Some minor revisions:

  • Line 13. (GSC) change with (GSCs) and check throughout the text;
  • In the introduction I suggest adding a part in which to mention studies in which other compounds both natural such as: saffron, crocetin, oleandrin, etc. (for example: Colapietro A., Mancini A., Vitale F., Martellucci S., Angelucci A., Llorens S., Mattei V., Gravina G.L., Alonso G.L., Festuccia C. Crocetin Extracted from Saffron Shows Antitumor Effects in Models of Human Glioblastoma. Int. J. Mol. Sci. 2020; 21:423. doi: 10.3390/ijms21020423; Colapietro A., Yang P., Rossetti A., Mancini A., Vitale F., Martellucci S., Conway T.L., Chakraborty S., Marampon F., Mattei V., et al. The Botanical Drug PBI-05204, a Supercritical CO2 Extract of Nerium Oleander, Inhibits Growth of Human Glioblastoma, Reduces Akt/mTOR Activities, and Modulates GSC Cell-Renewal Properties. Front. Pharmacol. 2020;11:552428. doi: 10.3389/fphar.2020.552428 and other) and synthetic compounds, have been used to modify the behavior of GBM cells and GSCs and their possible use as new therapeutic strategies.

  • Check the brackets of the bibliography in the text: replace ( ) with [ ].

  • Line 257 (fig. 5D) maybe it is 4D?

  • Line 468 change CO2 with CO2

  • Line 509 change and with at

  • Line 554 change small with Small

  • Line 557 change 200.000 with 200,000

  • Add a section for abbreviations
  • Completely review all references according to the journal's instructions: Author 1, A.B.; Author 2, C.D. Title. Abbrev Journal Year, Volume, page range.

Author Response

Reviewer #1: Comments and Suggestions for Authors

We thank the reviewer for the critical and helpful comments on our submitted manuscript. We have included a point-by-point answer to the reviewer’s suggestions.

Reviewer #1: Dear authors ...

I read this work with interest .... overall it seems to me well done and very clear with meaningful experiments and good figures and results. However, I suggest some changes to make it even more effective.

Some minor revisions:

Line 13. (GSC) change with (GSCs) and check throughout the text;

We thank the reviewer for this helpful hint. We changed the term GSC into GSCs throughout the text.

Reviewer #1: In the introduction I suggest adding a part in which to mention studies in which other compounds both natural such as: saffron, crocetin, oleandrin, etc. (for example: Colapietro A., Mancini A., Vitale F., Martellucci S., Angelucci A., Llorens S., Mattei V., Gravina G.L., Alonso G.L., Festuccia C. Crocetin Extracted from Saffron Shows Antitumor Effects in Models of Human Glioblastoma. Int. J. Mol. Sci. 2020; 21:423. doi: 10.3390/ijms21020423; Colapietro A., Yang P., Rossetti A., Mancini A., Vitale F., Martellucci S., Conway T.L., Chakraborty S., Marampon F., Mattei V., et al. The Botanical Drug PBI-05204, a Supercritical CO2 Extract of Nerium Oleander, Inhibits Growth of Human Glioblastoma, Reduces Akt/mTOR Activities, and Modulates GSC Cell-Renewal Properties. Front. Pharmacol. 2020;11:552428. doi: 10.3389/fphar.2020.552428 and other) and synthetic compounds, have been used to modify the behavior of GBM cells and GSCs and their possible use as new therapeutic strategies.

According to the reviewer’s suggestion, we included a part regarding studies in which both natural as well as synthetic compounds have been used to modify the behavior of GBM cells and GSCs.

The revised manuscript now reads:

Introduction, page 2, lines 51 to 56:

„Both natural and synthetic compounds have been evaluated for their potential to modify the behavior of GBM cells and GSCs, e.g. cannabinoids, resveratrol crocetin, the botanical drug PBI-05204, EGFR inhibitors, the PI3K/AKT small-molecule inhibitor A-443654, and many others [5-8]. However, despite knowing the mode of action of most of these com-pounds their possible clinical benefit, i.e. the impact on patient prognosis, is unclear to date.“  

Reviewer #1: Check the brackets of the bibliography in the text: replace ( ) with [ ].

We thank the reviewer for that comment and apologize for the mistake. We replaced ( ) with [ ].

Reviewer #1: Line 257 (fig. 5D) maybe it is 4D?

We apologize for the incorrect labelling. This fault has been removed in line 292 of the revised manuscript.

Reviewer #1: Line 468 change CO2 with CO2

We addressed the comment raised by the reviewer and corrected the term CO2

Reviewer #1: Line 509 change and with at

We thank the reviewer for the advice. We changed and with at in line 545 of the revised manuscript. 

Reviewer #1: Line 554 change small with Small

We agree with the reviewer and rephrased the respective line (line 590 of the revised manuscript).

Reviewer #1: Line 557 change 200.000 with 200,000

We thank the reviewer for this hint and corrected this fault in line 593 of the revised manuscript.

Reviewer #1: Add a section for abbreviations

We thank the reviewer for this valuable advice. We agree that a section for abbreviations might be helpful for the readership. We have therefore included an abbreviation chapter before starting with the Introduction.

Reviewer #1: Completely review all references according to the journal's instructions: Author 1, A.B.; Author 2, C.D. Title. Abbrev Journal Year, Volume, page range.

We thank the reviewer for the advice. We revised all references according to the journal´s instructions.

Reviewer 2 Report

The authors have to be congratulated for their effort since the topics of this article are very important and they are brought in a novel and inspiring way. However, there are some concerns, which should be addressed to assure the acceptance of this paper. Accordingly, I am afraid that the design of this manuscript is somewhat unusual. Therefore, I would suggest that the Introduction should be followed by Material and Methods and not the Results, which should be inserted after the Methods. The Discussion should be the last section of the manuscript, followed by a paragraph of clear Conclusions. The manuscript should end by few sentences addressing the study limitations. I would also try to avoid the term glioblastoma multiforme and replace it with glioblastoma only, since the previous term is rather obsolete, I believe.

If the author are willing to accept these suggestions, I am in favor of accepting the manuscript for publication with minor revisions.

Author Response

Reviewer #2: Comments and Suggestions for Authors

We thank the reviewer for the critical and helpful comments on our submitted manuscript. We have included a point-by-point answer to the reviewer’s suggestions.

Reviewer #2: The authors have to be congratulated for their effort since the topics of this article are very important and they are brought in a novel and inspiring way. However, there are some concerns, which should be addressed to assure the acceptance of this paper. Accordingly, I am afraid that the design of this manuscript is somewhat unusual. Therefore, I would suggest that the Introduction should be followed by Material and Methods and not the Results, which should be inserted after the Methods. The Discussion should be the last section of the manuscript, followed by a paragraph of clear Conclusions.

We agree that the design of this manuscript is somewhat unusual. Regarding the design of the manuscript, we followed the instruction of the International Journal of Molecular Sciences. In the International Journal of Molecular Sciences, the articles are usually structured as follows: 1. Introduction, 2. Results, 3. Discussion, and 4. Materials and Methods. Authors must use the Microsoft Word template or LaTeX template, specified by IJMS, to prepare their manuscript. 

Thus, we unfortunately cannot change the order of the sections.

Reviewer #2: The Discussion should be the last section of the manuscript, followed by a paragraph of clear Conclusions. The manuscript should end by few sentences addressing the study limitations.

We addressed this suggestion and included a paragraph of a final Conclusion after the Discussion section which also addresses study limitations.

The revised manuscript now reads:

Discussion section, page 16, lines 480 to 496:

„Conclusion and study limitations

Besides the described general relevance of PIM1 in cancer, our study shows for the first time that inhibition of PIM1 affects glioblastoma stem cell behavior and survival, suggesting PIM1 targeting as potential anti-glioblastoma therapy. However, our study also has limitations. We used two well-recognized cell lines (LN-18, U-87 MG) to analyze the role of PIM1. High genetic variation and rapid genetic diversification have been demonstrated for different cell lines, which may alter cancer cell line phenotype and drug response [73]. Thus, experiments with primary GBM cells freshly isolated from tumor tissue could complement the results of our study. Of note, the neurosphere assay used in our experiments is considered the gold standard for identifying the stem cell population in brain tissue. Neurosphere culturing is more relevant for modelling brain tumor biology than traditional 2D culture conditions, but important drawbacks of neurosphere assays are their inapplicability for detection of quiescent CSCs and the lack of the in vivo tumor environment [74-77]. To address the importance of tumor microenvironment for GSCs, future studies should include in vivo animal models with inoculation of PIM1 knockout GBM (neurosphere) cells, for example generated by the CRISPR/Cas9 technology, to further assess the impact of PIM1 on GBM pathogenesis.

Reviewer #2: I would also try to avoid the term glioblastoma multiforme and replace it with glioblastoma only, since the previous term is rather obsolete, I believe.

According to the suggestion of the reviewer, we replaced the term glioblastoma multiforme with glioblastoma only.

Reviewer #2: If the author are willing to accept these suggestions, I am in favor of accepting the manuscript for publication with minor revisions.

We thank the reviewer for his helpful advices.
